# Echo-ID: Smartphone Placement Region Identification for Context-Aware Computing

**DOI:** 10.3390/s23094302

**Published:** 2023-04-26

**Authors:** Xueting Jiang, Zhongning Zhao, Zhiyuan Li, Feng Hong

**Affiliations:** Department of Computer Science and Technology, Ocean University of China, Qingdao 266100, China; jiangxueting@stu.ouc.edu.cn (X.J.); zhaozhongning@stu.ouc.edu.cn (Z.Z.); lizhiyuan@stu.ouc.edu.cn (Z.L.)

**Keywords:** ultrasound sensing, region identification, deep learning

## Abstract

Region-function combinations are essential for smartphones to be intelligent and context-aware. The prerequisite for providing intelligent services is that the device can recognize the contextual region in which it resides. The existing region recognition schemes are mainly based on indoor positioning, which require pre-installed infrastructures or tedious calibration efforts or memory burden of precise locations. In addition, location classification recognition methods are limited by either their recognition granularity being too large (room-level) or too small (centimeter-level, requiring training data collection at multiple positions within the region), which constrains the applications of providing contextual awareness services based on region function combinations. In this paper, we propose a novel mobile system, called Echo-ID, that enables a phone to identify the region in which it resides without requiring any additional sensors or pre-installed infrastructure. Echo-ID applies Frequency Modulated Continuous Wave (FMCW) acoustic signals as its sensing medium which is transmitted and received by the speaker and microphones already available in common smartphones. The spatial relationships among the surrounding objects and the smartphone are extracted with a signal processing procedure. We further design a deep learning model to achieve accurate region identification, which calculate finer features inside the spatial relations, robust to phone placement uncertainty and environmental variation. Echo-ID requires users only to put their phone at two orthogonal angles for 8.5 s each inside a target region before use. We implement Echo-ID on the Android platform and evaluate it with Xiaomi 12 Pro and Honor-10 smartphones. Our experiments demonstrate that Echo-ID achieves an average accuracy of 94.6% for identifying five typical regions, with an improvement of 35.5% compared to EchoTag. The results confirm Echo-ID’s robustness and effectiveness for region identification.

## 1. Introduction

Context-aware computing refers to the ability of computing systems to automatically recognize and respond to the user’s scenario, providing intelligent and personalized services [1,2,3,4,5]. The global market size of context-aware services exceeded USD 36.3 billion in 2020 and is projected to reach over USD 318.6 billion by 2030 [6]. This growth is driven by the increasing adoption of mobile devices and the need for personalized and intelligent services.

With the prevalence of smartphones, there is a growing demand for smartphones to be more intelligent and context-aware. By capturing and utilizing contextual information, smartphones can automatically adjust their behavior based on the user’s current location, time of day, and activity level [1,2]. An example of context-aware computing with smartphones includes automatically activating the silent mode when a smartphone is placed on the nightstand to prevent disturbing the user’s sleep during the night. Similarly, when a smartphone is placed on a desk, it can enter a focus mode for a certain duration to ensure work quality, as shown in Figure 1. All these user scenarios require that the phone recognize the region where it resides and provide a specific function depending on the region. Note that users cannot place their phone on the exact location in the above scenarios, so context-aware services should be provided by identifying regions.

Despite the growing demand for a lightweight region-function combination for smartphones, such solutions are not yet pervasive. Traditionally, passive sensing solutions require pre-installed infrastructures or tedious calibration efforts. Solutions based on Wi-Fi [7], and Radio Frequency (RF) [8] require pre-installed equipment, such as wireless transmitters or beacons. Solutions that use indoor background lighting [9], sound [10,11], or mobile signal patterns [12] as fingerprints require laborious calibrations, which limits their practicality for personal use. Compared to other types of wireless sensing, acoustic sensing offers two advantages, (1) the widespread application of speakers and microphones in commercial electronic products, such as smartphones, smartwatches, and tablets, eliminates the need for additional hardware support; and (2) the lower propagation speed of sound in air compared to RF signal favors achieving higher distance measuring accuracy. Recently, active acoustic sensing solutions, such as EchoTag [13], enable phones to tag and remember locations by actively transmitting a sound signal with the phone’s speaker and sensing its reflections with the microphones. While EchoTag can provide a high 1cm resolution location-function combination, it requires users to place their phone in the exact location where they tagged it before, creating a new burden for users to remember and place their phone at the exact position and angle where it has been tagged.

To overcome these limitations, we propose Echo-ID, a low-cost region identification system that enables smartphones to identify regions and provide context-aware services without requiring any additional sensors or pre-installed infrastructure. Echo-ID actively generates an acoustic signature for target regions by using the phone’s speaker to transmit ultrasound and microphones to capture the reflections. Echo-ID requires users only to put their phone at two orthogonal angles for 8.5 s each inside a target region and bind a function with the region. The user can then place their phone freely in the target region and automatically obtain the context-aware functions in the future. Echo-ID only use the built-in speakers and microphones available in commodity smartphones and has low deployment and memory overhead for users.

However, it is non-trivial to identify regions for a smartphone only through ultrasound sensing. To ensure accuracy and robustness, we have to address two new challenges. (1) There is uncertainty in the phone placement inside a target region, which includes variations in position and angle. Asking users to place their phones at every possible position and angle to collect training data before using Echo-ID is impractical. (2) The reflective objects surrounding the target area may change over time. Therefore, Echo-ID must be able to accommodate placement uncertainty and environmental variations.

To address these challenges, Echo-ID applies Frequency Modulated Continuous Wave (FMCW) signals as a sensing medium and design a deep learning network to identify the target region. Echo-ID applies FMCW signals with carefully selected parameters. After removing the random delay by detecting the starting timestamp of the received sequence, the received signal is orthogonal demodulated to the baseband. Secondly, two types of coarse features are extracted from the baseband signals through Fast Fourier Transform (FFT) and Short Time Fourier Transform (STFT) guided by the FMCW ranging model, representing the coarse features of the spatial relationships among the surrounding reflective objects and the smartphone. Echo-ID’s deep learning model integrates residual network and attention mechanisms. The convolution operations inside residual network aim to dig out the invariant inside the coarse features, which is originally impacted by the phone placement uncertainty of position and angle. The attention mechanism may further focus on the core of spatial relationships among the surrounding reflective objects and the smartphone, reducing the impact of environmental changes for region identification. Though these two mechanisms, Echo-ID extracts finer features from the coarse features that emphasize the characteristics of the target region and are robust to phone placement uncertainty and environmental variation. Finally, the model identify the target region based on the finer features extracted.

We implement Echo-ID using the LibAS framework [14] on the Android platform and evaluate its performance on two smartphone models, Xiaomi 12 Pro and Honor-10, for identifying five typical regions, namely placing the phone on a nightstand, a kitchen operating table, a desk, a shelf, and a TV cabinet. The size of the Xiaomi 12 Pro device is 16.3 cm × 7.5 cm, and to allow users to place their phones easily and freely within the region, we set the target region at 35.6 cm × 35.6 cm, which is larger than twice of the phone’s longer edge. Theoretically, it would enhance Echo-ID’s ability to identify regions by obtaining training data from all conceivable positions and angles, but this imposes an impossible burden on users. Since Echo-ID represents the spatial relationship among surrounding objects in two-dimensional space under acoustic sensing, collecting training data by placing the smartphone at two orthogonal angles within the region may be possible to extract spatial features for classifying regions. Thus, the user only needs to place their phone at the region center’s two orthogonal angles for 8.5 s each to collect training data.

We use a Chirp of 42.6 ms as the sensing medium. For collecting training data, a Chirp is interleaved with a 42.6 ms idle slot, ensuring the elimination of reflection from distant objects in the received signals. Thus, we can obtain 100 spectrograms to train Echo-ID’s identification model from the sample collection of 8.5 s for each angle. To evaluate the impact of phone placement uncertainty, we evaluate the most varied 16 positions and 12 angles for phone placement inside each region. We also intentionally added or removed objects near the target regions to evaluate the system’s performance under environmental variations.

Our experiments demonstrate that Echo-ID achieved an average accuracy of 94.6% for identifying the five regions. We also thoroughly evaluated the system parameters that affect Echo-ID’s performance, including feature selection, residual module numbers, multi-angle combinations in the training set, smartphone model, permanence over time, and identification for 10 regions. The results confirm Echo-ID’s robustness and effectiveness for region identification.

The main contributions in Echo-ID are summarized as follows.
We propose Echo-ID, a low-cost region identification system for providing contextual services that exploits FMCW as a sensing medium. Echo-ID is not limited to placement position and angle, adapts to environmental changes, and without requiring any additional sensors or pre-installed infrastructure.Based on the extraction of spatial properties of surrounding reflective objects of the target region, we design a region identification model incorporating residual networks, attention mechanisms, and multi-layer perceptron to enable Echo-ID to achieve high accuracy in region recognition under the constraints of arbitrary placement of smartphones and dynamic nature of the surrounding environment.We implement Echo-ID on the Android platform and evaluate it with Xiaomi 12 Pro and Honor-10 smartphones, demonstrating its usability and robustness through extensive experiments.

The remainder of this paper is organized as follows: Section 2 describes the preliminary of FMCW signals and the ranging model. Section 3 provides a detailed description of Echo-ID design. Experimental details and the performance of Echo-ID are presented in Section 4. Section 5 discusses the related works. Finally, Section 6 summarizes the paper.

## 2. Preliminaries

In this section, we introduce the basics and the range model of FMCW signals used in acoustic sensing.

### 2.1. FMCW Fundamentals

FMCW is a radar technology that employs modulated continuous waves, where the transmitted signal is a sine wave whose frequency sweeps linearly over time, known as a Chirp signal. Compared to Continuous Wave (CW), the Chirp signal allows us to determine the absolute distance of a target and separate reflections from different distances. The transmitted signal $S_{T}\left(t\right)$ [15] can be obtained from Equation (Equation 1)
(1)$$S_{T}\left(t\right)=cos\left(\emptyset_{T}\left(t\right)\right)=cos\left(2\pi \left(f_{0}t+\frac{Bt^{2}}{2T_{s}}\right)\right)$$
where $f_{0}$ is the starting frequency, *B* is the bandwidth, $T_{s}$ is the sweep time. The signal frequency can be expressed as $f\left(t\right)=f_{0}+\frac{B}{T_{s}}t$. In Figure 2, the blue line represents $S_{T}\left(t\right)$.

### 2.2. Ranging Model

Chirps are commonly used in distance measurements between a target and a transceiver. In the case of acoustic sensing, a high-frequency Chirp is emitted from the speaker, and the signal reflected by surrounding objects is captured by the microphone. Consider a simple scenario where there is just a single reflection from the surrounding object. When the signal encounters a target at a distance of *R*, the reflected signal $S_{R}\left(t\right)$ is a time-delayed version of $S_{T}\left(t\right)$, which can be represented as
(2)$$S_{R}\left(t\right)=Acos\left(\emptyset_{T}\left(t-\tau \right)\right)=Acos\left(2\pi \left(f_{0}\left(t-\tau \right)+\frac{B{\left(t-\tau \right)}^{2}}{2T_{s}}\right)\right),$$
where *A* is the amplitude attenuation factor, and τ is the Time-of-Flight (ToF) of the signal in the air.

To calculate the distance between the transceiver and the object, the received signal needs to be demodulated by multiplying it with $cos\left(\emptyset_{T}\left(t\right)\right)$ and $-sin\left(\emptyset_{T}\left(t\right)\right)$. After applying the product-to-sum and product-to-difference conversion (i.e., $cos\alpha \cdot \cos\beta =\frac{1}{2}\left[cos\left(\alpha -\beta \right)+cos\left(\alpha +\beta \right)\right]$ and $cos\alpha \cdot \left(-\sin\beta \right)=\frac{1}{2}\left[\sin\left(\alpha -\beta \right)-\sin\left(\alpha +\beta \right)\right]$), the demodulated signal $S_{D1}\left(t\right)$ and $S_{D2}\left(t\right)$ are calculated according to Equations (Equation 3) and (Equation 4).
(3)$$\begin{array}{cc} S_{D1}\left(t\right) & =\frac{A}{2}\left(cos\left(2\pi \left(\frac{B\tau }{T_{s}}t+f_{0}\tau -\frac{B\tau^{2}}{2T_{s}}\right)\right)\right.\\ & +\left.cos\left(2\pi \left(-\frac{B}{T_{s}}t^{2}-2f_{0}t+\frac{B\tau }{T_{s}}t+f_{0}\tau -\frac{B\tau^{2}}{2T_{s}}\right)\right)\right), \end{array}$$
(4)$$\begin{array}{cc} S_{D2}\left(t\right) & =\frac{A}{2}\left(\sin\left(2\pi \left(\frac{B\tau }{T_{s}}t+f_{0}\tau -\frac{B\tau^{2}}{2T_{s}}\right)\right)\right.\\ & -\left.sin\left(2\pi \left(-\frac{B}{T_{s}}t^{2}-2f_{0}t+\frac{B\tau }{T_{s}}t+f_{0}\tau -\frac{B\tau^{2}}{2T_{s}}\right)\right)\right). \end{array}$$

The demodulated signal contains low-frequency and high-frequency components. The low-frequency component corresponds to the frequency difference between the received and transmitted signals, which is constant, labeled as $f_{d}=\frac{B\tau }{T_{s}}$. On the other hand, the high-frequency component represents the sum of their phases and varies with time, which can be filtered out by a low-pass filter. Figure 2 also shows an example of the received signals. The bottom part of Figure 2 shows the frequencies of the low- and high-frequency components of the demodulated signals. The low-frequency components of the demodulated signals $S_{D1}\left(t\right)$ and $S_{D2}\left(t\right)$ can be treated as the in-phase component, denoted as *I*, and the quadrature component, denoted as *Q*, respectively. The complex baseband signal $S_{B}\left(t\right)$ is generated from Equation (Equation 5).
(5)$$S_{B}\left(t\right)=\frac{A}{2}e^{-j\left(2\pi \left(\frac{B\tau }{T_{s}}t+f_{0}\tau -\frac{B\tau^{2}}{2T_{s}}\right)\right)}.$$

As shown in Figure 2, the baseband signal has a constant frequency, and, thus, $S_{B}\left(t\right)$ in Equation (Equation 5) can be simplified to Equation (Equation 6).
(6)$$S_{B}\left(t\right)=\frac{A}{2}e^{-j\left(2\pi f_{d}t+\varphi_{d}\right)},$$
where $f_{d}$ denotes the beat frequency, and $\varphi_{d}=2\pi f_{0}\tau -\frac{\pi B\tau^{2}}{T_{s}}$ is the phase. By calculating the beat frequency of the baseband signal, the propagation delay τ can be determined, and, thus, the distance between the device and the object can be calculated by $R=\frac{c\tau }{2}$, where *c* represents the speed of sound in air. Therefore, the radar using Chirp signals can find the distance between transceiver and the major reflected object, which is called the FMCW range model.

### 2.3. Multi-Target Range

In real-world scenarios, acoustic signals are reflected by multiple surrounding objects, leading to the superposition of reflected signals from different propagation paths at the microphone, called multipath effect. The received signal is a combination of signals from *L* paths, each with different delay and attenuation factors. The baseband signal can then be rewritten as Equation (Equation 7).
(7)$$S_{B}\left(t\right)=\sum_{l=1}^{L}\frac{A_{l}}{2}e^{-j\left(2\pi f_{d_{l}}t+\varphi_{d_{l}}\right)},$$
where $A_{l}$, $f_{d_{l}}$, $\varphi_{d_{l}}$ represents the attenuation factor, beat frequency, and phase of the signal received from the l-th path, respectively.

The time delays and beat frequencies of signals reflected from various distances create a one-to-one correspondence between propagation path length and baseband signal frequency. The spectrum of the baseband signal reveals the effect of reflections from different distances on the signal’s frequency, which can represent the spatial features of surrounding objects for a target region. The distance resolution of the FMCW system determines its ability to differentiate between two targets based on distance, with higher resolution allowing the radar to distinguish more closely spaced targets. When appropriate parameters for the FMCW acoustic signal are selected, a minimum resolvable distance of $\hat{d}=\frac{c}{2f_{s}}=$ 3.57 mm, which will be discussed in Section 4.2.

The acoustic characteristics of an environment is also represented by the uneven attenuation of the signal at different frequencies, as shown in Figure 3. This phenomenon can be attributed to three reasons, as reported in previous studies [16,17]. First, the imperfection in speakers and microphones will cause changes in the signals. Secondly, when sound reaches the surface of the reflected objects, the surface material may absorb signals at a particular frequency. Different materials have varying absorption frequencies. The combination of these two factors causes the received signal to amplify at certain frequencies and attenuate at others. Thirdly, the propagation of acoustic signals is subject to multipath effects when encountering reflections from surrounding objects, leading to different propagation paths and phase differences that impact the signal’s strength at different frequencies as discussed above. Therefore, Echo-ID may leverages the spectrum to recognize the region where the phone resides, because the spectrum presents the difference in the amplitudes on frequencies of the received signals.

## 3. System Design

This section first presents the design overview and then illustrates the design of each module.

### 3.1. System Overview

Echo-ID consists of four components, namely signal transmission, signal processing, coarse feature extraction, and a deep learning model, as illustrated in Figure 4. The signal transmission module transmits FMCW Chirps with appropriate parameters to obtain reflections from the surrounding objects. In the signal processing module, auto-correlation is employed to detect the Chirp start timestamp inside the received signals. Subsequently, orthogonal demodulation is performed to derive the baseband signal, followed by the elimination of the direct propagated component. Time-frequency domain analysis is used to extract spectrogram from the baseband signal, which serve as the coarse features for classification. Finally, the deep learning model employs a deep learning network that integrates convolution and residual networks with attention mechanisms to compute finer features and identify region.

### 3.2. Signal Transmission

Echo-ID transmits the Chirps to capture reflections from surrounding objects for region identification. The Chirp parameters include the frequency band and sweep time. As the sampling rate of the smartphone’s speakers is typically 48 kHz, the highest detectable sound frequency is limited to 24 kHz according to the Nyquist-Shannon sampling theorem. To avoid audible noise and ensure compatibility with most commercial smartphones, the scanning frequency band of 18 kHz–22 kHz is selected since humans can hear sounds within the range of 20 Hz to 18 kHz [15].

The sweep time is closely related to the Signal-to-Noise Ratio (SNR) of the received signal. Although a longer sweep time can increase the reflecting range and frequency resolution, it also consumes more energy and may introduce interference from distant objects. Since Echo-ID is designed to recognize specific regions, reflected signals from distant objects far from the target area are not necessary. Hence, a sweep time of 42.6 ms is chosen for each Chirp sequence, resulting in 2048 samples for each sweep, to obtain sufficient frequency resolution and avoid interference caused by distant reflections. The time-domain and spectrogram of the transmitted Chirp sequence are depicted in Figure 5.

### 3.3. Signal Processing

This module performs three operations, including removing the random delay by auto-correlation, orthogonal demodulation, and direct propagated component removal. To identify the start of the received Chirp, we leverage the strong correlation property of the Chirps. The Chirp is a CAZAC sequence [18] with a constant envelope and zero auto-correlation characteristics, making it an ideal candidate for detecting the start with auto-correlation. By correlating the transmitted Chirp with the received signal, the timestamp corresponding to the maximum correlation value marks the start of the received Chirp. An example of the auto-correlation result is illustrated in Figure 6, where the peak time (42.6 ms) indicates the start of the received Chirp.

After obtaining the received Chirp, demodulation is performed to transform it into the baseband. The orthogonal demodulation process is illustrated in Figure 7, where the received signal is multiplied by the Chirp and passed through a low-pass filter to separate the in-phase component I and quadrature component Q. Echo-ID employs an equiripple low-pass filter, which is a linear-phase finite impulse response filter, to retains the phase spectrum of the received signals. The filter’s pass-band cut-off frequency is set to 4 kHz, fully preserving the effective signal.

To avoid significant interference from directly propagated signals that travel from the speaker to the microphones, it is necessary to measure the direct propagation path and remove the low-frequency components that correspond to this distance, according to the FMCW range model. This is because the distances of the directly propagated signals are shorter than those of the reflected signals.

### 3.4. Coarse Feature Extraction

Context-aware computing refers to the ability of computing systems to automatically recognize and respond to the user’s scenario, providing intelligent and personalized services. An example of context-aware computing with smartphones includes automatically activating the silent mode when a smartphone is placed on the nightstand to prevent disturbing the user’s sleep during the night. Similarly, when a smartphone is placed on a desk or shelf, we hope it can enter a focus mode for a certain duration to ensure work quality. When a smartphone is on a TV cabinet, the phone volume is automatically turned up to avoid missing calls and messages. When a smartphone is placed on a kitchen operating table, the sound is transmitted to the speaker for playback, or a predefined timer is started when it is near a device, such as a microwave oven.

All these user scenarios require that the phone recognize the region where it resides and provide a specific function depending on the region. Additionally, the surrounding objects within the kitchen and desk scenes change frequently, which helps to verify the validity of Echo-ID. These scenarios reflect how users can benefit from region–function combinations of context-aware services. Therefore, we positioned the smartphone in these five regions for experiments, on a nightstand, a kitchen operating table, a desk, a shelf, and a TV cabinet, as shown in Figure 8.

Since the reflections from multiple objects are combined within the received signals, it is crucial to extract features from the baseband signals. The frequency spectrum of the baseband signal corresponds to different reflection paths from objects at various distances, as discussed in the FMCW range model, and can be obtained through FFT. Figure 9 displays the frequency spectrum of five regions, exhibiting differences, particularly at their peak values. The distributions of peak frequencies represent the major reflected signal paths, corresponding to the locations of surrounding reflective objects. For example, the two peaks in the amplitude of the nightstand marks the frequency of 102 Hz and 177 Hz, which labels two reflective objects, i.e., the glasses and lamp, at a distance of about 20.4 cm and 35.4 cm, respectively.

Echo-ID requires a feature that portrays the spatial relationships between the surrounding objects and the smartphone. According to the FMCW range model, the peak in the amplitude spectrum labels a reflected path of the surrounding object with the peak corresponding to the distance between the object and the smartphone. However, FFT does not account for the arrival times of the reflected signals along multiple reflection paths, so it majorly treats a reflective object as a particle.

Considering that actual objects have a continuous reflective area, we apply STFT on aggregated signal sequences to extract spatial features. Through the fine-grained movement of the time window (0.1 ms), STFT can capture a more detailed description of the multi-path reflected signals by the same object surface. In other words, an extreme point of the frequency corresponding to the amplitude spectrum will be a hot zone in the spectrogram around the same frequency with a time length representing the difference between the shortest and longest reflected path. Moreover, other transforms, such as Wavelet Transform (WT), Hilbert-Huang Transform (HHT) [19], and Discrete Tchebichef polynomials (DTPs) [20], may also provide different features from the received acoustic sequences, but since they do not represent the distance, like Fourier transform and STFT, we will try it in future work.

Because Echo-ID achieves region identification based on the absolute distance from the object around the device to the microphone and its reflective surface size, the signal variation caused by objects around the device is more important for Echo-ID. Therefore, we only focus on the frequency from 0 to 500 Hz in the spectrograms, corresponding to a range of 90 cm around the device. Figure 10 shows variations in spectrograms among five regions, especially in the frequency and time corresponding to the hot zones. The distribution of the highlight areas represents the spatiotemporal distribution of the arrival of the reflected signals reflected by the surrounding objects. In Figure 10a, the two hot zones also mark the two reflective objects (the lamp and glasses). The area difference in the two hot zones further indicates different sizes of the continuous reflective area of two objects, which can serve more clues about the surrounding objects. We call the spectrum and spectrogram features as coarse features, because the noises are still buried inside. In our evaluation, we will compare the spectrum and spectrogram feature for region identification.

### 3.5. Deep Learning Model

Echo-ID leverages deep learning networks to extract finer features from the coarse features and recognize region in which a user places their smartphone, taking into account the variations of each displacement. For example, a user may place their phone on a desk while working, but the precise location of the phone may vary within a given region each time. To accommodate these variations in device placement, Echo-ID uses deep learning to capture the unique characteristics of each region. The network architecture is depicted in Figure 11.

To extract fine-grained features from spectrograms, the learning network employs a convolution neural network (CNN) [21,22] and a residual network (ResNet) [23] as its backbone, which take 256×256 spectrograms of amplitude and phase as input. The CNN’s convolution operation extracts local features while preserving spatial information, and pooling operations reduce the dimension of the feature map, enhancing the model’s computational efficiency. As the CNN depth increases, the model’s spatial receptive field expands, capturing global spatial features across the frequency band and time domain. However, excessively deep networks can cause issues such as gradient vanishing or exploding, as well as smaller gradient updates for shallow CNN layers, which can increase training difficulty. To address these issues, ResNet incorporates a shortcut path from input to output, allowing the model to learn the residual between the input and output directly during back propagation. By using a deep learning network for feature extraction, fine-grained features inside the spectrograms can be effectively extracted.

To emphasize the differences among these fine-grained features, a lightweight convolution block attention module (CBAM) [24] is integrated into the residual network. CBAM adds attention in the channel and spatial dimensions by performing pooling operations while maintaining the feature map’s channel or spatial dimension unchanged. A Multi-Layer Perceptron (MLP) generates attention values for each channel or spatial dimension, and the attention values for both the channel and spatial dimensions are multiplied and added to the feature map. By calculating the attention weight for each channel and spatial dimension, the feature map is weighted, enhancing key features, suppressing noise and irrelevant features, and enabling the model to focus more on specific features in a certain time and frequency range. CBAM can visualize the model’s attention regions using the classification activation map (CAM) [25], which will be presented in the evaluation section.

Finally, the extracted fine-grained features are concatenated into one feature vector, which is then classified using a MLP. Table 1 provides the parameter configurations for each module as well as the corresponding input and output dimensions.

Figure 12a–e shows the spectrogram of five placement modes in the kitchen operating table region and Figure 12f shows the concatenated features from the well-trained model. When the positions and angles of the phone and environmental changes, the deep learning model can still extract the fine-grained features and, thus, gives similar feature values before the classifier. Figure 12g,h shows that the CBAM helps the model pay more attention to specific features in a certain time and frequency range. We input the spectrogram in Figure 10 to the model and show the concatenated features from the well-trained model in Figure 13a. Compared to Figure 13f, the deep learning model is able to magnify the features difference in distinct regions. Figure 13b,c also demonstrate that our model can enhance key features in distinct regions.

In principle, obtaining training data from all conceivable positions and angles would enhance Echo-ID’s ability to identify regions, but this would also impose a burden on users. Since Echo-ID represents the spatial relationship among surrounding objects in two-dimensional space under acoustic sensing, collecting training data by placing the smartphone at two orthogonal angles at the region center is sufficient. Thus, the user only needs to place their phone at the region center’s two orthogonal angles for 8.5 s each to collect training data. During the 8.5 s, a Chirp of 42.6 ms is interleaved with a 42.6 ms idle slot, ensuring the elimination of reflection from distant objects in the received signals. The deep learning model design compensates for position and angle differences through convolution operations and accommodates the surrounding changes through the attention mechanism in region identification. As Section 4.4 demonstrates, collecting training data at more than two orthogonal angles only slightly improves region identification accuracy.

## 4. Experiment and Evaluation

In this section, we conduct comprehensive experiments to demonstrate the effectiveness of Echo-ID.

### 4.1. Experiment

#### 4.1.1. Settings

We implemented Echo-ID on the Android platform based on LibAS [14] and conducted experiments to evaluate its performance using Xiaomi 12 Pro and Honor-10 smartphones in five different regions, as shown in Figure 8. The target regions are 35.6 cm × 35.6 cm, which accommodates the size of the Xiaomi 12 Pro device (16.3 cm × 7.5 cm), allowing users to place their phones easily and freely within the region. The deep learning model is completely implemented by PyTorch with a RTX 3060 Ti GPU. The network is trained using Adam with a learning rate = $1\times 10^{-4}$.

#### 4.1.2. Dataset

Echo-ID uses the speaker at the bottom to send out FMCW signals and the two microphones at the top and bottom to receive signals. During the experiment, individuals in the environment are not restricted in their activities, such as typing on a keyboard or walking.

To reduce the burden for users before using the Echo-ID, we collect the training set by placing the smartphones inside the target region with two orthogonal angles of 0° and 90°, as shown in Figure 14a, after which the system can recognize the region in the future. Echo-ID requires only 8.5 s at each angle to collect 100 FMCW traces of 42.6 ms with a 42.6 ms idle slot inserted between two Chirps to eliminate the distant reflection on the subsequent trace. The total time cost of training data collection is 17 s, which is easy for users to accept. The total size of the training set obtained for the five regions is 1000 traces.

To fully evaluate the impact of phone placement uncertainty on region identification, we select 16 different positions and 12 angles, as illustrated in Figure 14b,c. Additionally, we simulate environmental variations by adding or removing reflective objects, such as cups, bowls, and books, to assess Echo-ID’s adaptability.

To evaluate the effect of different coarse features extracted from the received signal on the recognition capability, we compare different features from the received signals and baseband signals, including received signal spectrum, baseband signal spectrum, baseband signal spectrogram, and spectrum and spectrogram of baseband signal.

To fully assess the impact of different numbers of residual blocks on the network classification performance, we evaluate the network identification accuracy with 2, 3, 4, and 5 residual blocks.

To determine whether the training set should include data from multiple angles, we train models with data collected from 1 to 6 different angles at the center position and tested them with data from other angles and the changed environment.

To verify the effect of differences in microphones and speakers used by different smartphones, we evaluate the usability of Echo-ID with Xiaomi 12 Pro and Honor-10.

To evaluate the permanence of region identification, the experiments last for 12 weeks across the summer, fall, and winter seasons. We collect 56,000 traces every week, resulting in a total of 672,000 traces. The traces in the first week are used for training and the remaining traces are used for testing.

At the end of this section, we choose five regions where smartphones are placed less frequently to test the applicability of Echo-ID. The dataset is available at https://github.com/JiangXueting1225/Echo-ID, accessed on 27 March 2023.

### 4.2. Identification Accuracy

The identification accuracy of Echo-ID for five regions is shown in Figure 15a. Considering noise, imperfect hardware, and accommodating environmental changes in practice, it is difficult for this solution to achieve 100% recognition accuracy. Moreover, the different scenario may require different acceptable accuracy according to their special region-combination. For example, the nightstand scenario may require high accuracy in order to ensure the quality of the user’s sleep. The results in Figure 15 show that the nightstand has the highest accuracy of 100% to ensure that the user’s preset mode, such as mute, can be activated. This indicates that reflecting objects in this environment were highly discriminable, resulting in the least path interference. The kitchen operating table and TV cabinet have lower identification accuracy, possibly due to the presence of multiple objects with continuous reflecting surfaces, like the walls. In the desk region, users usually need to turn on focus mode. However, the desk region is partially recognized as the nightstand, likely due to the concentrated objects on one side. The lowest accuracy of 88% exists for the desk region, attributed to the high degree of dynamic factors in the office/lab environment that affected the extraction of time-frequency domain features. When the user finds that the mode is not turned on, Echo-ID can be performed again. The whole process only takes only 42.6 ms, which is shorter than manually turning on functions such as switching to focus mode. The overall accuracy of 94.6% ensures the probability of correct identification for continuous 12 times is still over 50%. This indicates that Echo-ID can automatically identify usage regions in a user’s daily life and adapt to various placement conditions and variations in the surrounding environment. And this only requires the user to collect training data of 17 s in five regions each. Hence, the identification accuracy in terms of user experience is acceptable.

Furthermore, we compare Echo-ID with EchoTag in Figure 15b which uses the received signal spectrum as features, and Support Vector Machines (SVM) for region classification. The identification accuracy of EchoTag was 69.8%, lower than Echo-ID, especially for the nightstand region. EchoTag does not account for placement ambiguity and environmental variations, whereas Echo-ID demonstrated compatibility with phone placement positions, angles, and environmental variations by applying residual networks with attention modules to accommodate the differences inside one region. Moreover, we combine Echo-ID features with a SVM classifier, whose results are shown in Figure 15c with an average accuracy of 75%. The result is higher than EchoTag, confirming that the Echo-ID’s features are superior to represent the region’s characteristics.

To evaluate the compatibility of Echo-ID, we present the identification results for various phone placement positions, angles, and environmental variations in Figure 16. Figure 16a indicates that positions closer to the center have higher accuracy, while positions further away have lower accuracy. Nevertheless, the identification accuracy of 90.2% is sufficient to demonstrate compatibility with phone placement positions. Figure 16b reveals that accuracy is higher for specific angles (30°, 60°, 120°, and 330°), likely due to their similarity in energy distribution with training angles, but decreases for other angles. Accuracy is relatively high for 180° and 270° due to the use of two pairs of microphones at the top and bottom of the device to receive the signals. When the phone is placed at 180° and 270°, the dual microphones receive signals similar to those received when the device is placed at 0° and 90°. Figure 16c displays identification results for environmental variations, with the nightstand having the highest accuracy due to that most of its reflection paths are from walls and lamps. The movement of cups does not significantly affect the features, while the accuracy of the desk is the lowest since the main reflection object after removing the cup and tissue box is the computer monitor. Nevertheless, the accuracy of 85% for the desk is sufficient to indicate that Echo-ID can be compatible with dynamic environmental variations.

EchoTag achieves centimeter-level spatial position recognition. If the user is willing to clearly mark the location where they place their phone, EchoTag will provide contextual services with more spatial granularity than Echo-ID. For example, drawing the outline of the phone with a pen at the target location and carefully placing the phone in that position with the same placement angle each time. When the two locations are spaced 5 cm apart on a desk, Echo-Tag can recognize and activate two different contextual services for each location. In addition, EchoTag puts so much effort into centimeter-level location discrimination that it deviates when there is a small change of surrounding objects positions.

The major concern here is whether the user is willing to tag out every exact phone location and put their phone carefully at these positions with the same placement angles. Moreover, the users rarely require two contextual services when their phone locations are only 5 cm different. In contrast, Echo-ID aims to provide region-function combinations for users, giving them more freedom to place their phones. The target to provide context services is a region with an area with each edge twice than the longer edge of the phone. In addition, Echo-ID puts no restrictions on the phone placement angle, and it can adapt to small changes in the surrounding environment over time. To collect training data, Echo-ID only requires the user to place their phone at the two orthogonal angles of the region center for 8.5 s each. The spectrogram feature and the designed deep learning network model make Echo-ID robust against phone placement uncertainty and environmental changes.

Figure 17 shows the attention areas of the magnitude and phase spectrogram for 5 regions, deduced from the last layer of CBAM. The low-frequency portion of the amplitude variation within 20.5 cm from the smartphone microphone was the focus of the kitchen operating table and TV cabinet, while the higher frequency amplitude variation in the distant position was of more concern to the nightstand, desk, and shelf. Regarding the phase domain attention areas, the nightstand concentrated on the phase variation at around 26 ms and 42 ms, while the kitchen operating table paid attention to the phase variation throughout the period from 8.3 ms to 40 ms. The desk only showed attentiveness to the phase variation within 8 ms, while the kitchen operating table and TV cabinet were attentive to the phase variation in three different periods. The variation of the attention areas of magnitude and phase across different regions suggests that Echo-ID can distinguish these 5 regions.

### 4.3. Acoustic Feature

The ability of region identification is directly affected by the features extracted from the received signals. The features compared are shown in Table 2, and Figure 18a presents the results of region identification using different features. The highest classification accuracy is achieved using the spectrogram feature for all regions. This is because it expresses the spatial distribution of the environment’s reflective objects through the spectrum and also captures the signal delay differences of different reflection paths through time-domain window division.

The spectrum feature of the baseband signal achieves the second-highest classification accuracy, with similar results for the nightstand and slightly lower results for the other four regions. This validates the effectiveness of the FMCW’s spatial ranging principle, which reflects the spatial distribution of reflective objects in the environment to a certain extent. The spectrum feature’s classification accuracy is lower than that of the spectrogram feature because the entire spectrum of the reception period is directly input to the identification network, whereas the spectrogram provides multi-level spectra divided by the continuous time windows, increasing the spatial distribution details for different periods.

The joint feature of the baseband achieves lower results than the first two features because the network structure becomes more complex, leading to overfitting due to the small size of the training dataset. Using the spectrum feature of the received signal without demodulation is also less accurate than the first two features, as the dominant feature in the original signal spectrum is the modulation frequency band, submerging the spatial distribution spectrum performance. The accuracy of using the received signal’s spectrum feature for identification is lower than that of EchoTag, particularly in the kitchen operating table, shelf, and TV cabinet regions, because the region identification network in this paper cannot obtain spatial distribution information from the spectrum of the received signal, making it difficult to distinguish environments with more reflective objects.

The nightstand environment consistently achieves perfect accuracy (100%) in both the spectrum and spectrogram characteristics of the baseband signal across all regions. This is due to the high discrimination of reflection distribution, which results in the least amount of confusion in spatial distribution for signals reaching the receiving device, and, therefore, the region can be correctly identified as long as the spatial distribution features are effectively extracted. The shelf and kitchen operating table regions have lower accuracy in the frequency spectrum characteristics due to the continuous reflective surfaces of objects confusing spatial distribution. Although time-domain segmentation improves the recognition rate of these regions, the desk region has the lowest identification rate equivalent to the frequency spectrum characteristics due to its high degree of dynamics.

Figure 18b displays the comparison of the average accuracy of each characteristic, and the identification accuracy of the spectrogram feature is the highest at 94.6%. This is approximately 12% higher than the frequency spectrum characteristic, which ranks second, indicating the efficacy of Echo-ID’s use of the time-frequency spectrum characteristic for region identification.

### 4.4. Number of Residual Blocks

The classification performance of the network is affected by the number of residual blocks used. We evaluate the influence of different numbers of residual blocks on Echo-ID’s recognition outcomes and assess network identification accuracy with 2, 3, 4, and 5 residual blocks, shown in Figure 19.

The highest identification accuracy for each region is achieved with three residual blocks, resulting in an average identification accuracy of 94.6%. Using 2, 4, or 5 residual blocks results in a decrease in accuracy by 18.8%, 12.8%, and 16.2%, respectively. This is because when only two residual blocks are utilized, the model may not be able to capture the complexity of the input features, and the network cannot fully learn the features for classification from the training data, resulting in underfitting. When there are too many residual blocks, the model may be too complex, leading to overfitting and reduced generalization ability on the test set.

Regardless of the number of residual blocks used, the nightstand consistently achieved the highest accuracy in all regions, indicating that the reflectivity distribution on the nightstand is the most distinguishable and the features are more distinct. Conversely, the identification accuracy of other regions decreased significantly with different numbers of residual blocks. The kitchen operating table, desk, and shelf regions had particularly low recognition accuracy due to the continuous reflective surfaces, such as walls, nearby. Figure 19b displays the average identification accuracy for different numbers of residual blocks, indicating that the network with three residual blocks used in Echo-ID is effective for region identification.

### 4.5. Size of Training Set

To determine whether the training set should include data from multiple angles, we train models with data collected from 1 to 6 different angles at the center position, including {0°}, {0°, 90°}, {0°, 120°, 240°}, {0°, 90°, 150°, 300°}, {0°, 90°, 150°, 210°, 300°}, and {0°, 60°, 120°, 180°, 240°, 300°}, and test them with data from other angles and the changed environment. The identification accuracy reaches 94.6% when only two angles were used for training, as shown in Figure 20a, and slightly improves of about 0.9% with additional training data. When the training data are increasing to six angles, the identification accuracy only increases by 0.9%, but it adds three times burden for data collection.

Moreover, we train models with data collected from 1 to 8 different position combinations with two orthogonal angles, including position {6}, {1, 6}, {1, 6, 11}, {1, 6, 11, 16}, {1, 4, 6, 11, 16}, {1, 4, 6, 11, 13, 16}, {1, 4, 6, 7, 11, 13, 16}, and {1, 4, 6, 7, 10, 11, 13, 16}. Figure 20b shows that adding the training data of more positions with two orthogonal angles also does not increase the identification accuracy greatly, and the accuracy only increases by 1.0% with training data of eight positions.

These results can be attributed to the fact that all placements in the target region can be represented by two orthogonal angles at the region center as the spatial base. Therefore, to reduce the burden on users under the premise of ensuring accuracy, Echo-ID only requires users to place their phones horizontally and vertically at the center of the target region for 8.5 s each to collect training data.

### 4.6. Mobile Phone Model

To assess the robustness of Echo-ID across different phone models, we also collect the signal traces using Honor-10 smartphones. We divide all the collected traces into four categories, Dataset I uses the samples from two angles of the Xiaomi 12 Pro for training and all other samples for testing; Dataset II uses the samples from two angles of the Honor-10 for training and all other samples for testing; Dataset III uses data samples from two angles of the Xiaomi 12 Pro for training and all samples from the Honor-10 for testing; and Dataset IV uses a mixture of samples from both Honor-10 and Xiaomi 12 Pro for training and testing.

As shown in Figure 20c, Echo-ID achieves accuracy rates of 94.6% for Dataset I, 90.2% for Dataset II, 71.8% for Dataset III, and 79.6% for Dataset IV. This first confirms that Echo-ID can also be used for Honor-10 smartphones. It also suggests that the highest accuracy is achieved when training and testing are performed on the same phone model. The accuracy of mixed training and testing using two different smartphones is the second highest, while the lowest accuracy is achieved when training on one phone and testing on another. Therefore, when deploying Echo-ID on a new smartphone, it is better to collect training data specific to the new model. We will study whether transfer learning will help accommodate different phone models in the future.

### 4.7. Permanence

To examine the permanence of region identification, we divide the six-week traces according to their weeks, using only the 8.5 s traces for two orthogonal angles in the first week for training and the remaining data for testing. Figure 20d illustrates the identification accuracy of the five regions for each week, showing that the identification accuracy is stable over time.

### 4.8. Ten Regions Identification

To validate the effectiveness of Echo-ID, we expand the region variable to ten. The deployment of the remaining five regions is illustrated in Figure 21. Figure 22 displays the confusion matrix for the identification of ten indoor regions using Echo-ID, with region index referencing Table 3. The identification accuracy for all ten regions is above 69%, with corridors achieving the highest accuracy (98%) due to their spaciousness and lack of reflective objects, which distinguish them from other regions. Classrooms have the lowest accuracy (69%) because their layout is similar to that of bedside tables, desks, and washbasins, all of which have a prominent reflective object on one side. More than half of the regions have an accuracy of over 85%, and the overall accuracy for all ten regions is 82.9%. These findings suggest that Echo-ID can effectively perceive regions.

## 5. Related Works

Currently, the main target related to the region identification of Echo-ID is the indoor positioning research. Global Positioning System (GPS) sensors in mobile devices has been widely used for high-precision outdoor positioning [26]. However, accurate indoor location sensing remains challenging due to the inability of GPS signals to penetrate buildings. In recent years, research efforts have emerged to address indoor location using various signal processing techniques, including Global System for Mobile Communications (GSM) [12], FM radio [27], Wi-Fi [7], and acoustic signals [13]. This section provides an overview of existing indoor location systems, which are categorized into three types based on their perceptual modalities, dedicated equipment, RF signal, and acoustic signal.

### 5.1. Dedicated Equipment

Specialized devices are previously used for indoor location sensing. For instance, WALRUS [10] uses ultrasonic transmitters and receivers to locate targets in a room by measuring propagation time and Angle of Arrival (AoA). Wang et al. [8] track employees’ indoor locations by attaching battery-powered radio transmitters to their work badges. Addlesee et al. [7] use various sensors and devices to collect environmental samples and automate responses to environmental changes through machine learning and intelligent algorithms. Liu et al. [28] implement smartphone location sensing by placing ultrasonic transmitters with unique identity tags indoors.

The dedicated equipment requires pre-installation of beacons throughout the environment, with higher installation and maintenance costs for infrastructure deployment, imposing an additional burden on the user.

### 5.2. RF Signal

The use of RF signals for indoor location sensing relies on the existing wireless infrastructure, such as GSM and Wi-Fi, to determine the location of mobile devices. This is achieved by measuring the Received Signal Strength Indicator (RSSI) from cellular or Wi-Fi stations with known locations. Haeberlen et al. [29] collected RSSI measurements from access points in a large-scale 802.11 Wi-Fi network to achieve location sensing. Hightower et al. [12] established a geographic information database using Wi-Fi, cellular, and geomagnetic signals to identify the user’s location. Horus [30] utilized multiple Wi-Fi receivers to determine the wireless device’s position through the analysis of parameters, such as RSSI, arrival time, and AoA. Vasisht et al. [31] and Boney et al. [32] implemented indoor location sensing using RSSI from a single Wi-Fi access point.

The localization accuracy of such schemes depends on the density of cellular or Wi-Fi stations in the environment, and the RSSI is susceptible to fluctuations due to multi-path fading in complex indoor environments.

### 5.3. Acoustic

Acoustic sensing has been widely adopted in various smart sensing scenarios, such as identity verification [33,34,35,36,37,38,39,40], health monitoring [41,42,43,44,45], input interfaces [46,47,48,49,50], keystroke inference [51,52], and gesture recognition [53,54,55]. Compared to other types of wireless sensing, acoustic sensing offers two advantages, (1) the widespread application of speakers and microphones in commercial electronic products, such as smartphones, smartwatches, and tablets eliminates the need for additional hardware support; and (2) the lower propagation speed of sound in air compared to RF signal favors achieving higher distance measuring accuracy.

Passive acoustic location sensing captures environmental sounds as fingerprints for indoor positioning [9,11,56]. However, these solutions are susceptible to the influence of environmental noise, such as music or conversation. In addition, passive acoustic location solutions are greatly affected when the environment changes, for example, Batphone [11] notes that when the Heating, Ventilation, and Air Conditioning (HVAC) is turned off for maintenance, the characteristics of the background sound can change significantly, which can interfere with the accuracy of locating single or multiple sound sources. To address this issue, some studies [57,58] have proposed using microphone arrays to locate sound sources more precisely.

Active acoustic location sensing emits pre-designed acoustic signals through the speaker and capture their echoes using the microphone to estimate the device’s position [16,59,60,61,62,63]. Tachikawa et al. [59] and IndoLabel [63] combine active and passive localization techniques to obtain impulse response features or MFCC coefficients by transmitting a Chirp sequence, and they combine features from multiple other sensors to achieve room-level accuracy for indoor localization. The closest to Echo-ID is EchoTag [13], which successfully employed active acoustic sensing to tag and remember indoor locations by enriching acoustic features with different delays Chirp signals. Recent studies [64,65] have proposed using active acoustic sensing to estimate room shapes.

The existing acoustic-based location sensing systems mentioned above consider the entire reflected signal as an acoustic fingerprint and necessitate the sound source to follow a predefined path or require additional microphone arrays to be installed in the room. Furthermore, these systems are restricted by the placement location and angle of smartphones, posing a challenge for users to remember and place their phones precisely, and their performance degrades considerably when environmental conditions change. In contrast, Echo-ID overcomes the limitations of existing acoustic-based location sensing systems by utilizing FMCW chirps to generate acoustic features for regions with the built-in speakers and microphones, and a deep learning model that is capable of handling placement uncertainty and environmental variations.

## 6. Conclusions and Future Work

The paper proposes Echo-ID, a low-cost region identification system that uses the built-in sensors available in commodity smartphones to identify regions and provide context-aware services without requiring additional sensors or pre-installed infrastructure. Echo-ID applies FMCW signals as a sensing medium and designs a deep learning network to recognize the target region. We implement Echo-ID on the Android platform and evaluate it with Xiaomi 12 Pro and Honor-10 smartphones, demonstrating its usability and robustness through extensive experiments. Echo-ID achieves an average accuracy of 94.6% for identifying the five regions which only requires users placing their phones in each target region at two orthogonal angles for 8.5 s.

We only explore the feasibility and robustness of region identification with STFT to extract the spatial features. We will investigate other types of discrete transforms for spatial feature extraction in the future. Moreover, the users need to place their phones on purpose at the region center’s two orthogonal angles for 8.5 s each to collect training data. Our future study will propose an online learning scheme that lets Echo-ID learn the placement regions automatically and only requires the user to decide whether make a combination between the learned region and contextual services.

## Figures and Tables

**Figure 1 sensors-23-04302-f001:**
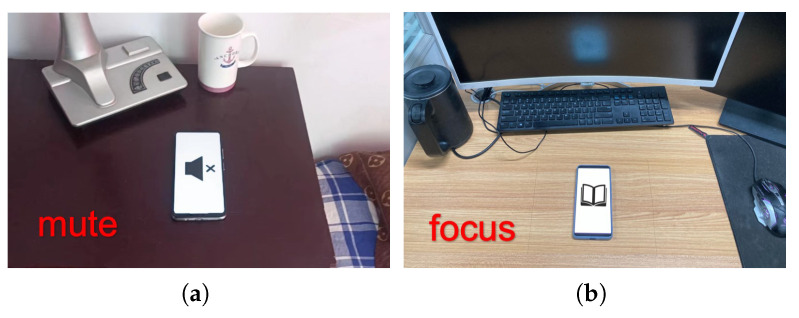
Examples of Echo-ID user scenarios. (**a**) Automatic activation of silent mode when placed on a nightstand and (**b**) automatic initiation of focus mode when placed on a work desk.

**Figure 2 sensors-23-04302-f002:**
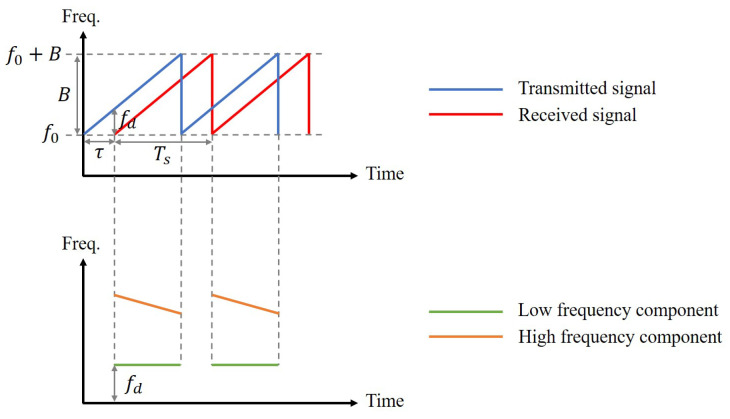
Sketch of transmission and reception of FMCW signal.

**Figure 3 sensors-23-04302-f003:**
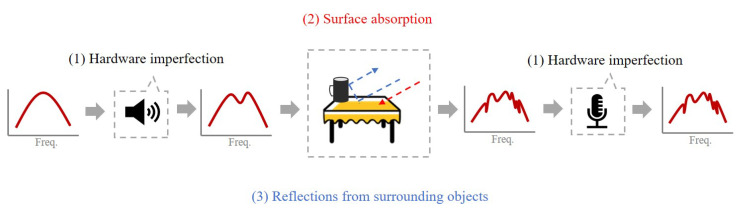
Uneven signal attenuation at different frequencies and the corresponding reasons.

**Figure 4 sensors-23-04302-f004:**
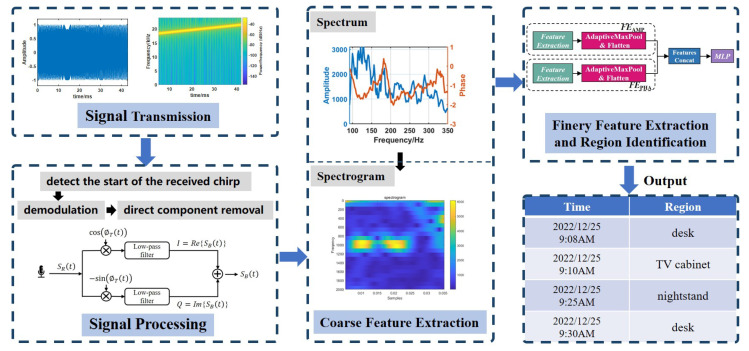
System overview of Echo−ID.

**Figure 5 sensors-23-04302-f005:**
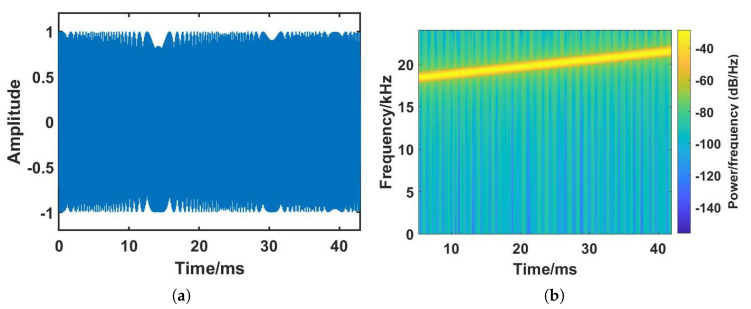
Transmitted Chirp. (**a**) Time domain and (**b**) Spectrogram.

**Figure 6 sensors-23-04302-f006:**
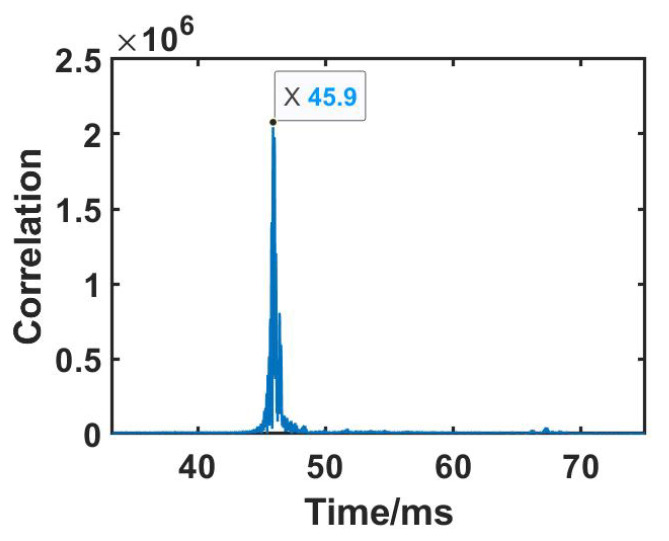
The start time of the received Chirp.

**Figure 7 sensors-23-04302-f007:**
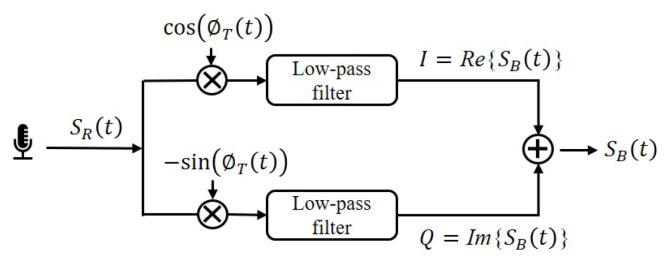
Orthogonal demodulation.

**Figure 8 sensors-23-04302-f008:**
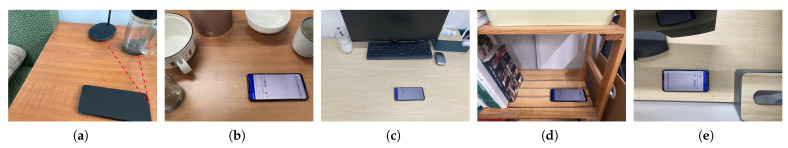
Five typical regions. (**a**) Nightstand. (**b**) Kitchen operating table. (**c**) Desk. (**d**) Shelf. (**e**) TV cabinet.

**Figure 9 sensors-23-04302-f009:**
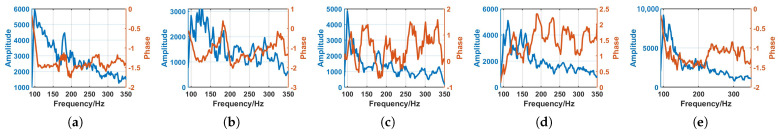
Frequency spectrum of five regions. (**a**) Nightstand. (**b**) Kitchen operating table. (**c**) Desk. (**d**) Shelf. (**e**) TV cabinet.

**Figure 10 sensors-23-04302-f010:**
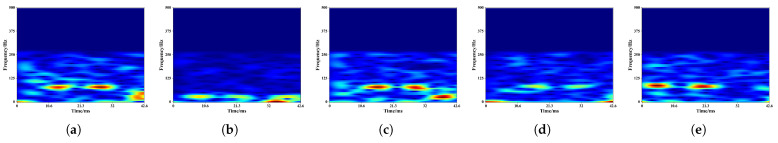
Spectrogram of five regions. (**a**) Nightstand. (**b**) Kitchen operating table. (**c**) Desk. (**d**) Shelf. (**e**) TV cabinet.

**Figure 11 sensors-23-04302-f011:**
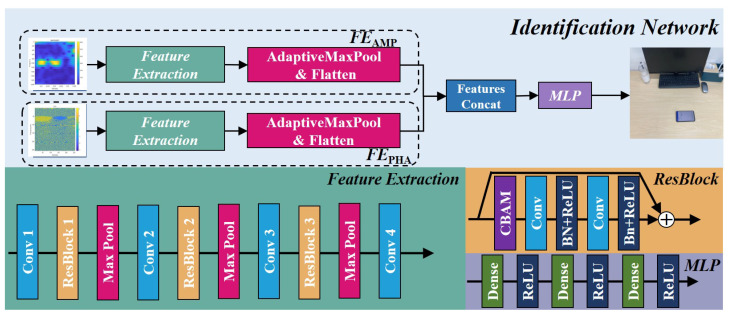
Finer feature extraction and region identification network structure.

**Figure 12 sensors-23-04302-f012:**
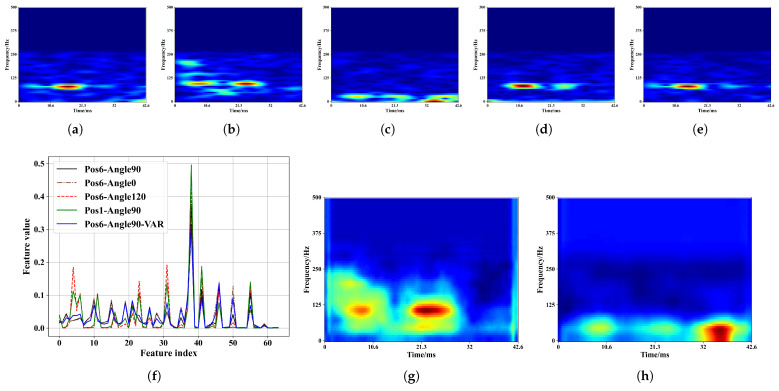
Spectrogram of five placement modes in the kitchen operating table region and visualization of features. (**a**) Nightstand. (**b**) Kitchen operating table. (**c**) Desk. (**d**) Shelf. (**e**) TV cabinet. (**f**) Concatenated feature values of different placement modes. (**g**) Attention area of (**b**). (**h**) Attention area of (**c**).

**Figure 13 sensors-23-04302-f013:**
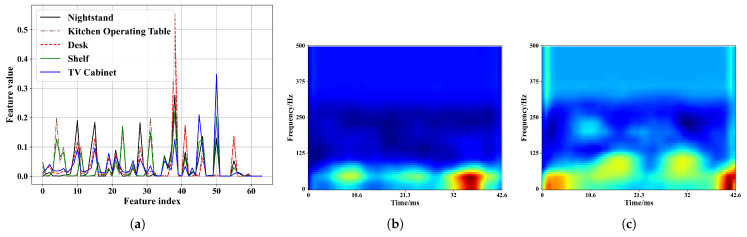
Visualization of features after input Figure 10 into the model. (**a**) Concatenated feature values of different regions. (**b**) Attention area of Figure 10b. (**c**) Attention area of Figure 10d.

**Figure 14 sensors-23-04302-f014:**
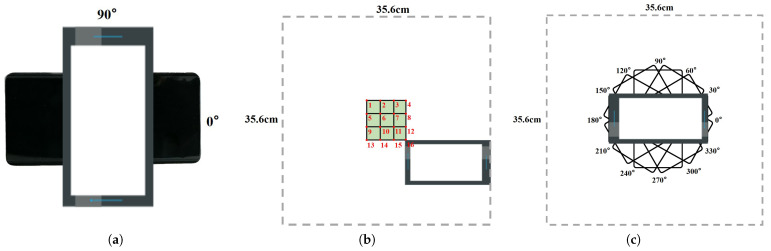
Phone placement for experimental data collection. (**a**) Collecting training traces at 2 orthogonal angles. (**b**) Collecting test traces at 16 positions. (**c**) Collecting test traces at 12 angles.

**Figure 15 sensors-23-04302-f015:**
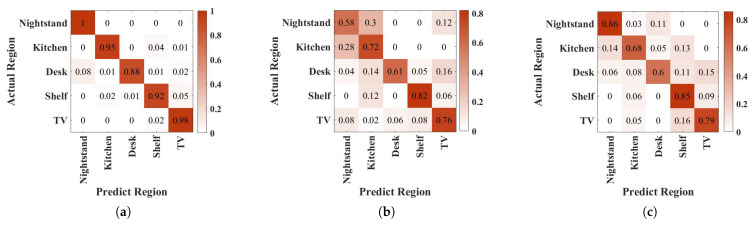
Confusion matrices. (**a**) Echo-ID. (**b**) EchoTag. (**c**) Echo-ID’s coarse feature + SVM.

**Figure 16 sensors-23-04302-f016:**
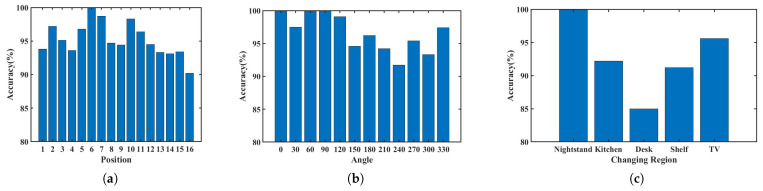
Identification accuracy (**a**) of 16 positions (**b**) of 12 angles (**c**) with environmental changes.

**Figure 17 sensors-23-04302-f017:**
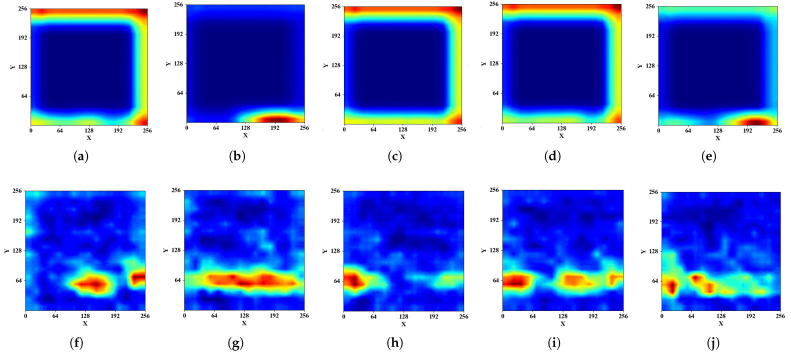
Attention results: amplitude of (**a**) Nightstand. (**b**) Kitchen operating table. (**c**) Desk. (**d**) Shelf. (**e**) TV cabinet; phase of (**f**) Nightstand. (**g**) Kitchen operating table. (**h**) Desk. (**i**) Shelf. (**j**) TV cabinet.

**Figure 18 sensors-23-04302-f018:**
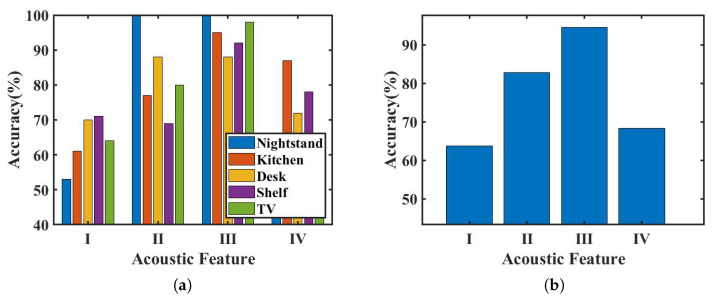
Identification accuracy of different features. (**a**) Accuracy of each region. (**b**) Average accuracy.

**Figure 19 sensors-23-04302-f019:**
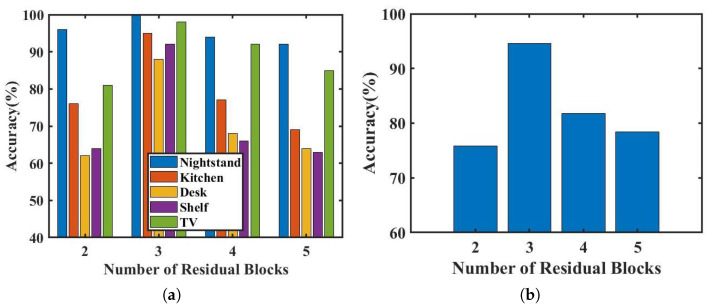
Identification accuracy of different numbers of residual blocks. (**a**) Accuracy of each region. (**b**) Average accuracy.

**Figure 20 sensors-23-04302-f020:**
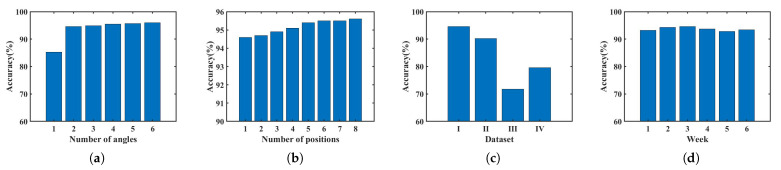
Identification accuracy. (**a**) Using traces collected from multiple angles for training. (**b**) Using traces collected from multiple positions for training. (**c**) Different phone model. (**d**) Permanence.

**Figure 21 sensors-23-04302-f021:**
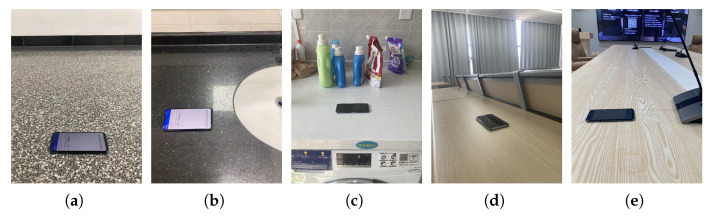
Experimental scenarios for other five regions. (**a**) Corridor. (**b**) Washbasin. (**c**) Washing machine. (**d**) Classroom. (**e**) Conference room.

**Figure 22 sensors-23-04302-f022:**
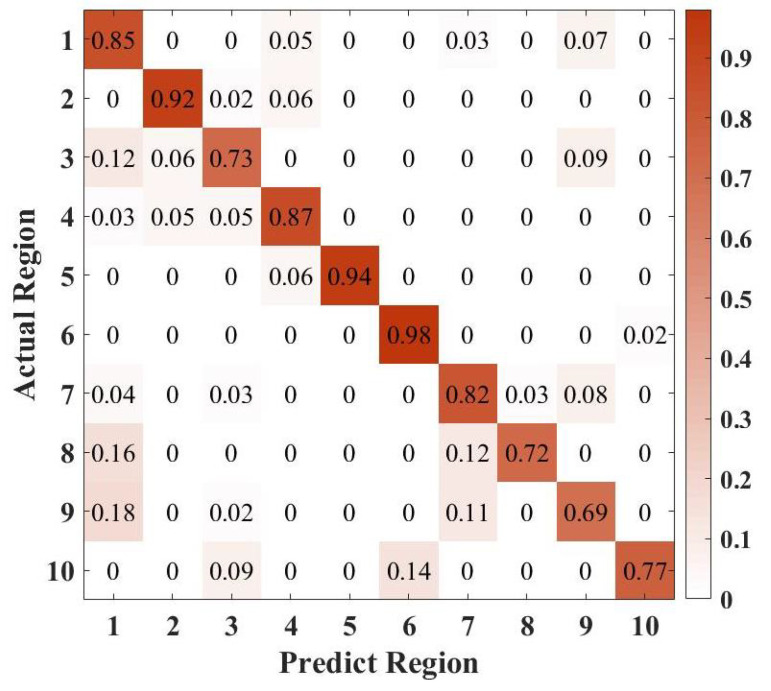
Confusion matrix of ten regions identification.

**Table 1 sensors-23-04302-t001:** Configuring the parameters and data flow dimensions for every module within the finer feature extraction and region identification network.

Model Name	Input Dimension	Parameter Settings	Output Dimension
Conv 1	1 × 256 × 256	32 × 3 × 3	32 × 256 × 256
ResBlock 1	32 × 256 × 256	32 × 3 × 3	32 × 256 × 256
MaxPooling	32 × 256 × 256	2 × 2	32 × 128 × 128
Conv 2	32 × 128 × 128	32 × 3 × 3	32 × 128 × 128
ResBlock 2	32 × 128 × 128	32 × 3 × 3	32 × 128 × 128
MaxPooling	32 × 128 × 128	2 × 2	32 × 64 × 64
Conv 3	32 × 64 × 64	32 × 3 × 3	32 × 64 × 64
ResBlock 3	32 × 64 × 64	32 × 3 × 3	32 × 64 × 64
MaxPooling	32 × 64 × 64	2 × 2	32 × 32 × 32
Conv 4	32 × 32 × 32	32 × 3 × 3	32 × 32 × 32
AdaptiveMaxPool + Flatten	32 × 32 × 32	-	32
Feature Fusion	32 (Amplitude feature) 32 (Phase feature)	-	64
MLP	64	64 × 512 512 × 1024 1024 × 512 512 × 5	5 (Probability of classifying 5 regions)
Total number of parameters	180M		

**Table 2 sensors-23-04302-t002:** Acoustic Feature.

Feature I	Feature II	Feature III	Feature IV
received signal spectrum	baseband signal spectrum	baseband signal spectrogram	spectrum and spectrogram of baseband signal

**Table 3 sensors-23-04302-t003:** Index of 10 regions.

Index	Region	Index	Region
1	nightstand	6	corridor
2	kitchen operating table	7	washbasin
3	desk	8	washing machine
4	shelf	9	classroom
5	TV cabinet	10	wonference room

## Data Availability

The data presented in this study are available in Section 4.1.2.

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
