# Peer review of "Echo-ID: Smartphone Placement Region Identification for Context-Aware Computing"

_sensors, 2023, doi:10.3390/s23094302_

Round 1

Reviewer 1 Report

Echo-ID: Smartphone Placement Region Identification for Context-Aware Computing

In abstract FMCW is used, without defining it at first instance.

Introduction about Mobile devices and their context awareness based on regions is explained well with suitable references 1 to 13.

Lines 88 to 97, needs some explanation why these settings are taken? Elaborate more to offer better understandings.

Line 122, why again now FMCW defined, its defined earlier

Rest of the work is presented wonderfully

It’s a good work can be accepted with such small changes

Kindly check the figures for some better clarity, Fig 17,

Check all used short forms must be defined only once throughout except abstract

Useful work for research community.

Can be improved a bit

Reviewer 2 Report

Title: Echo-ID: Smartphone Placement Region Identification for Context-Aware Computing

Summary:

In this work, a mobile system is proposed to allow phones to identify their current region without the need for additional sensors or pre-installed infrastructure. The proposed system is called Echo-ID which applies frequency-modulated continuous wave (FMCW) s as a sensing medium.

The manuscript is interesting; however, the following comments need to be addressed carefully:

Abstract:

- - - - - - - - -

1 - Define FMCW first then use.

2- remove extra “to” in the sentence “… robust to to phone …” - - > “robust to phone” .

3 – problem statement need to be more elaborated .

4 – Improvement ratio need to be included at the end of the abstract. The improvement ratio should be between the proposed and existing works .

Introduction Section :

- - - - - - - - - - - - - - - - - -

5 – The contributions list need to be rewritten such that the second point should be merged with the first point .

6 – some CNN based applications need to be included in this section such as : a) doi: 10.3390/electronics11193084 , b) doi: 10.1109/DeSE54285.2021.9719469, and c) doi: 10.1016/j.rineng.2023.100969.

Preliminaries Section :

- - - - - - - - - - - - - - - - - - - -

7 – “… is the bandwidth, Ts is the sweep time, and the signal … ” should read “… is the bandwidth, Ts is the sweep time. The signal … ”

8 – Include reference(s) for equations 1, and 2 .

9 – Rearrange the flow sequence in Figure 3 for more readability .

10 – Check for equations that require references .

System Design Section :

- - - - - - - - - - - - - - - - - - - - -

11 – change the limit of the y-axis of Figure 10 from 0 to 500 Hz.

12 - change the limit of the y-axis of Figure 12 (a-e) from 0 to 500 Hz.

13 – The use of STFT is justified in “However, Fourier transform does not account for the arrival times of the reflected 257 signals along multiple reflection paths, which can be addressed using STFT [18]”. However, there are other types of discrete transforms than can be used such as 10.1002/cpe.7311 . Include justification for other type of transforms.

Experiment and Evaluation Section :

- - - - - - - - - - - - - - - - - - - - - - - - - - - - -

14 – The dataset need to be available .

15 – For Figure 16, the identification accuracy need to be from 80 to 100 to increase the readability .

16 – For Figure 18, the identification accuracy need to be from 40 to 100 to increase the readability .

17 – For Figure 19 and 20, the identification accuracy need to be from 60 to 100 to increase the readability .

Conclusion Section :

- - - - - - - - - - - - - - - - -

18 – Include limitations and future works .

References :

- - - - - - - - - - - - - -

19 – Update the references from 2021 and 2022 literature .

- - - - - - - - - - - - - - - - - - - - - - - - - - - - - - - - - - - - - - - - - - - - - - - - - - - - - - - - - - - - - - - - - - - - - - - - - - - - - - - - - - - - - - - - - - - - - - - - - - - - - - - - - - - - - - - - - - - - - - - - - - - - - - - - - - - - - - - - - - - - - - - - - - - - - - - - - - - - - - - - - - - - - - - - - - - - - - - - - - - - - - - - - - - - - - - - - - - - - - - - - - - - - - - - - - - - - - - - - - - - - - - - - - - - - - - - - - - - - - - - - - - - - - - - - - - - - - - - - - - - - - - - - - - - - - - - - - - - - - - - - - - - - - - - - - - - - - - - - - - -

The language has some minor error and typos.

Reviewer 3 Report

This paper proposes and investigates a new approach to region-function combinations for use in context-aware computing problems. For the most part, I think the paper is fairly strong. The underlying study is worthwhile and much of the paper is well written. I do have a few reservations, however, which I believe the authors should be invited to address:

·         The authors should present (briefly) a more explicit argument about why ‘acoustic signals’-type approaches are, in general, seen as so promising for context-aware computing problems. The abstract does already hint at the answer, but in my view the Introduction should state this in slightly more detail.

·         In my view, the authors should compare their Echo-ID approach with the prior work on EchoTag in a more nuanced way. Are there trade-offs to be made, in terms of technology and user experience, when choosing between the two approaches?

·         The introduction should also explain *why* a target region size of 35.6cm x 35.6cm was chosen, rather than just stating that this was the case.

·         I was less convinced about the strengths of section 2 than the rest of the paper. Is this section really needed? What is distinctive here to the Echo-ID work? This section currently reads more like a textbook than an empirical paper. I’d suggest either deleting the section or signposting more directly what is distinctive and new in the content.

·         Throughout section 3, but especially in section 3.4, I think we need more nuanced discussion of the use cases that are being chosen as the basis of the investigation. Saying that certain regions are “typical”, as is done on line 249, is not really enough explanation.

·         It would also be useful to reflect on what trade-offs are regarded as acceptable, in terms of positions and angles for calibration, in terms of accuracy versus user experience (e.g., around lines 317 onwards) and user burdens (around line 347). Why are these regarded as acceptable?

·         Similarly, are there established thresholds being used for acceptable identification accuracy in terms of user experience? (line 378 onwards).

·         Section 5 would benefit from establishing more directly what the present work *contributes* to the prior literature. In other words, the idea would be to relate the present proposal to these related works and establish the points of commonality and difference more directly.

Round 2

Reviewer 2 Report

Title: Echo-ID: Smartphone Placement Region Identification for Context-Aware Computing

Summary:

In this work, a mobile system is proposed to allow phones to identify their current region without the need for additional sensors or pre-installed infrastructure. The proposed system is called Echo-ID which applies frequency-modulated continuous wave (FMCW) s as a sensing medium.

In the revised manuscript, the authors have addressed the raised comments.

There are a few minor errors that can be addressed during the proof read process.